# Mechanical-scan-free multicolor super-resolution imaging with diffractive spot array illumination

Ning Xu [1], Sarah E. Bohndiek [2,3], Zexing Li[4], Cilong Zhang [1] & Qiaofeng Tan [1]✉

Point-scanning microscopy approaches are transforming super-resolution imaging. Despite achieving parallel high-speed imaging using multifocal techniques, efficient multicolor imaging methods with high-quality illumination are currently lacking. In this paper, we present for the first time Mechanical-scan-free multiColor Super-resolution Microscopy (MCoSM) with spot array illumination, which enables mechanical-scan-free super-resolution imaging with adjustable resolution and a good effective field-of-view based on spatial light modulators. Through 100–2,500 s super-resolution spot illumination with different effective fields of view for imaging, we demonstrate the adjustable capacity of MCoSM. MCoSM extends existing spectral imaging capabilities through a time-sharing process involving different color illumination with phase-shift scanning while retaining the spatial flexibility of super-resolution imaging with diffractive spot array illumination. To demonstrate the prospects of MCoSM, we perform four-color imaging of fluorescent beads at high resolution. MCoSM provides a versatile platform for studying molecular interactions in complex samples at the nanoscale level.

Point-scanning super-resolution microscopy, such as confocal microscopy (CM)[1,2], structured illumination microscopy (SIM)[3], and stimulated emission depletion (STED) microscopy[4], has become indispensable for investigating nanoscale biological structures and interactions in biomedical research. Among the various techniques, CM physically provides a $\sqrt{2}$ times resolution at best with one spot via a focused beam and pinhole detection. SIM increases the cutoff frequency of the optical transfer function to obtain theoretical double-resolution, achieving widefield imaging at relatively low phototoxicity[5]. The performance of STED microscopy scales with the illumination intensity; moreover, although such approaches constrain the choice of dyes and must balance the spatial resolution and phototoxicity, which enables cellular imaging with a resolution as low as 20 nm[6].

For multicolor super-resolution imaging, the use of distinct fluorescent labels enables researchers to study spatial relationships between different molecular processes within cells[7]. Traditional point-scanning super-resolution microscopy can be challenging when applied for multicolor imaging due to the long data acquisition time, which normally exacerbates photobleaching. In addition, movement of the microscope stage typically requires mechanical adjustments, which can perturb the sample during refocusing, although this issue can sometimes be mitigated with remote focusing[8,9]. Notably, galvanometer scanners[10] and scan lenses[11,12] are commonly used to achieve relative displacements between the sample and illumination system, which inevitably increases in size. These issues have led to difficulty in achieving multicolor fluorescence imaging across wide fields of view with nanometer-scale precision in biological samples.

[1]State Key Laboratory of Precision Measurement Technology and Instruments, Department of Precision Instrument, Tsinghua University, Beijing 100084, China. [2]Department of Physics, Cavendish Laboratory, University of Cambridge, JJ Thomson Avenue, Cambridge CB3 0HE, UK. [3]Cancer Research UK Cambridge Institute, University of Cambridge, Robinson Way, Cambridge CB2 0RE, UK. [4]Department of Pure Mathematics and Mathematical Statistics, University of Cambridge, Wilberforce Road, Cambridge CB3 0WB, UK. ✉e-mail: tanqf@mail.tsinghua.edu.cn

To reduce the data acquisition time, multifocality has been explored in super-resolution microscopy to extend the total effective field-of-view (FoV)[13]. If the conventional single spot is replaced with a sufficiently large spot array, the data acquisition time can be significantly reduced, as a large spot array requires only a few sampling steps. For a 50 × 50 spot array with a resolution of 70% of the Airy spot (0.7 Airy), the amount of data collected at each position is increased by ~1750-fold, with a 30% improvement in the lateral resolution compared with a single Airy spot illumination. Therefore, the spot array can reduce scan times by 2–3 orders of magnitude compared to the acquisition time of single spot illumination, while improving the resolution. For a Fourier transform system, the maximum size of the diffraction field without aliasing in the output plane is restricted by $\lambda f / \Delta p$, where $f$ is the focal length of the objective and $\Delta p$ is the sampling interval of the diffractive optical element (DOE)[14]. Generally, the spot array does not fully use the entire diffractive field, and the effective FoV is determined by the number of spot arrays.

Nonetheless, creating super-resolution spot arrays beyond the diffraction limit remains challenging. Initially proposed by Toraldo di Francia[15], the focusing of a single super-resolution spot beyond the Abbe-Rayleigh diffraction limit can be realized by wavefront shaping[16–19], computational deconvolution[20] or some combination of these methods[21,22]. Flexible phase distributions can be used to generate a range of modulated intensity distributions by harnessing the power and affordability of diffractive optics, leading to the development of new illumination strategies for optical microscopy and nanoscopy; moreover, mass production is feasible via nanoprinting, with the potential cost per element reduced to only a few dollars[23,24]. In particular, metasurfaces can be used to develop miniaturized devices with high resolution[25–27]. Super-resolution spot arrays can achieve high resolution with diffractive optics but at the expense of extremely low light efficiency due to high intensity sidelobes, which limits their applications in nanoscale imaging[28–30]. In our previous work, we experimentally generated 3 × 3 super-resolution spot arrays with diffractive optical elements[31]; however, achieving a higher effective FoV by increasing the number of spots with higher resolution in state-of-the-art microscopy is challenging in practice.

Although the development of new hardware and brighter, more photostable, fluorophores have led to the creation of faster and more sensitive nanoscale imaging techniques, there are still inherent speed limitations in fluorescence super-resolution microscopy. Typically, multicolor images are routinely obtained by employing a time-sharing process to acquire the expected organelle information with different incident wavelengths applied sequentially[32,33]. Notably, the wavelength-selective phase-shift technique has been adopted in digital holography[34] and three-dimensional imaging[35,36] to replace mechanically moving components. Various organelles can be imaged via phase-shift scanning with high efficiency without mechanically perturbing the sample.

To achieve rapid multicolor nanoscale fluorescence imaging with high effective FoV, adjustable resolution, without mechanical perturbations of the sample, in this paper, we design and build a Mechanical-scan-free multiColor Super-resolution Microscopy (MCoSM) system. MCoSM achieves mechanical-scan-free imaging with adjustable diffractive spot arrays using spatial light modulators (SLMs). As a proof-of-concept study, we imaged fluorescent beads and biological tissues using the MCoSM system, where the maximum scanning range and minimum step size were 210.4 μm and 79.1 nm on both the $x$ and $y$ axes, respectively. The MCoSM system can experimentally achieve four-color imaging with a resolution >96 nm using a 10 × 10 spot array, with a resolution of 52% of the Airy spot (0.52 Airy), outperforming the prior state-of-the-art[31] and achieving multicolor phase-shift scanning by superimposing and rapidly modulating the designed phase distribution on the SLM. The proposed system has significant advantages in controlling the beam modulation and scanning process with the same SLM, enabling high precision and rapid imaging at the given wavelengths. As every phase distribution corresponds to a specific wavelength and spatial position, scanning with the SLM allows the spot array to be moved any location within the study region, thereby realizing mechanical-scan-free imaging.

## Results

### Principles of MCoSM

In MCoSM system, illumination is provided by super-resolution spot arrays, with adjustable numbers of spots and spot sizes generated by SLMs. Multicolor imaging is achieved by automatically alternating multimodal acquisitions sequentially (four independent channels) and phase-shift scanning. Successive excitation in different visible bands is achieved by lasers with different wavelengths (Fig. 1a). As the number of labels increases, the incident wavelengths can be correspondingly increased. Every wavelength is split into two by a beam splitter (BS), and the modulated transmitted beam is reflected by the SLM. After being reflected at the BS, the beam illuminates the entrance pupil plane of the objective. Then, the spot array is generated in the sample plane, located at the focal plane of the objective, to illuminate the target object. The power and time delay can be adjusted to control the excitation windows in a largely independent manner. The incident wavelength $\lambda_n (n \in N^*)$ can be adjusted to achieve multicolor excitation processes and multicolor phase-shift scanning with the excitation of the corresponding chromophores. This strategy provides efficient and sequential excitation of different fluorescent labels with the phase-shift scanning process. In other words, the spatial resolution and spectral channels in the MCoSM system can be easily altered by switching the parameters of the illumination spot array to meet the desired requirements (see Methods). The reconstruction process can be summarized as 'determine the center position of the spots and extract the central intensity'. Thus, we improved the resolution by physical means without using deconvolution or other processes (see Data processing).

Super-resolution information is not directly obtained by the imaging system and thus must be reconstructed. The intensity of the MCoSM system at the detector can be analytically expressed as[37]

$$I(\xi,\eta) = \iint p_{\text{illu}}(u,v)t(u,v)p_{\text{imag}}(\xi - u, \eta - v)dudv, \qquad (1)$$

where $p_{\text{illu}}(u,v)$ and $p_{\text{imag}}(\xi,\eta)$ are the point spread functions (PSFs) of the illumination system and imaging system, respectively, and $t(u,v)$ is the intensity transmittance function of the sample. Previously, we showed that super-resolution information can be obtained from the convolution of the imaging system with 3 × 3 or 5 × 5 spot array incidence patterns via numerical simulations when the numerical aperture (NA) of the illumination system is equal to that of the imaging objective and the distance between the adjacent spot centers is twice the spot size[31]. To demonstrate that the super-resolution information can also be obtained when the number of spots is extended to $N^2$ ($N \in N^*$), we first mathematically proved that $N \times N$ super-resolution spot array illumination enables super-resolution imaging (the distance between the adjacent spot center is twice the size of the spot) (see Supplementary Information 1).

The MCoSM system obtains scans of the sample with multiple excitations, and the resulting fluorescence at each scan position is mapped to the corresponding pixels in the image. It is often assumed that the speed of single spot illumination methods ($N = 1$ in the MCoSM system) can be improved by increasing the scan speed; however, the resulting decrease in the per-pixel dwell time reduces the quality of the overall signal and degrades the image's signal-to-noise ratio (SNR)[38]. Increasing the illumination intensity compensates for this effect but can lead to increased photodamage and photobleaching (and at high intensities, these processes can scale nonlinearly with intensity). The

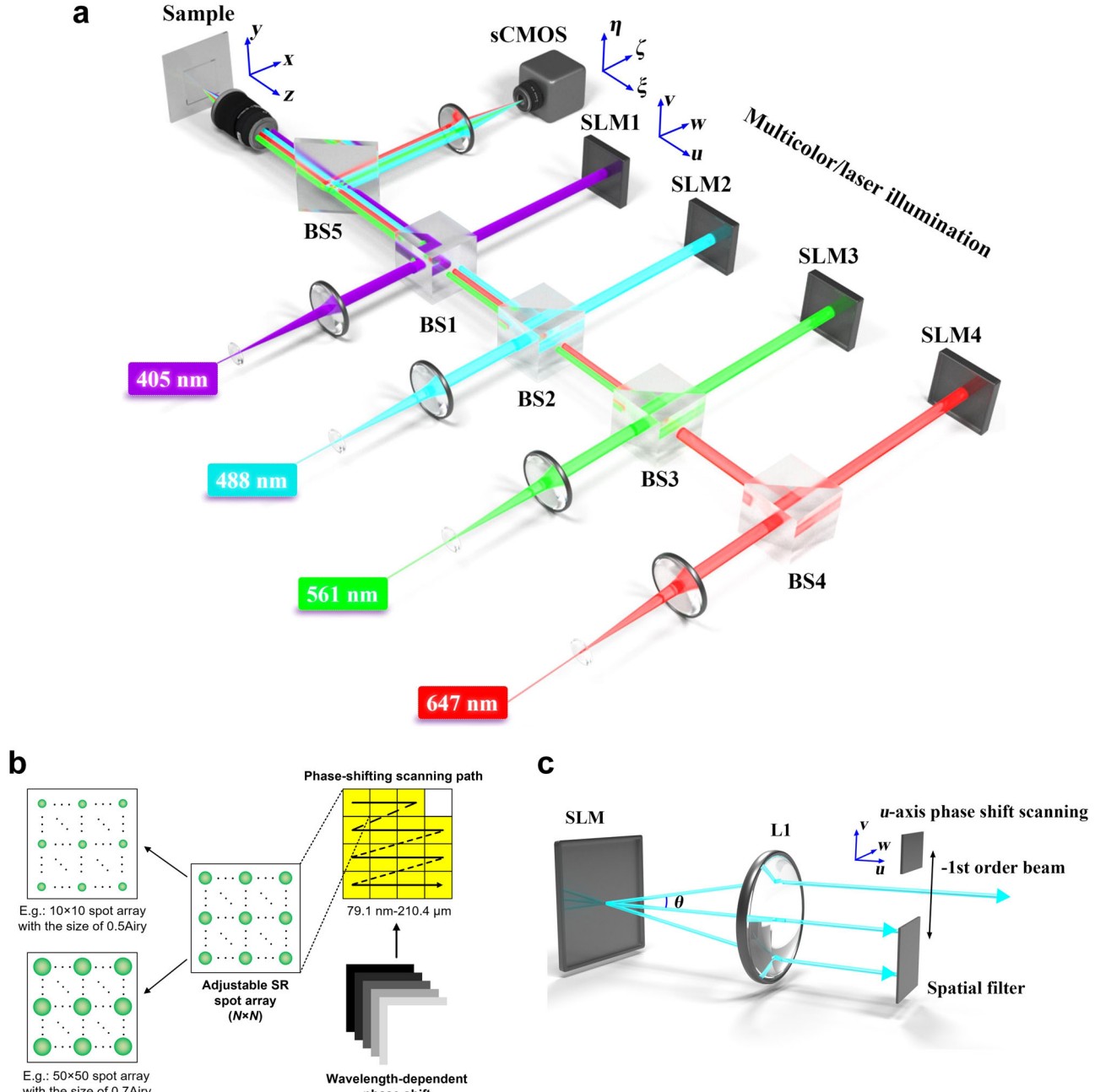

**Fig. 1 | Optical design of the MCoSM system and illumination phase-shift scanning path. a** The system is assembled using 4 SLMs with plane wave incidence to generate $N \times N$ super-resolution spot(s) on the sample, and an imaging detector, to relay super-resolution information. **b** Phase-shift scanning path of the spot array, which is realized by adding different phase shifts to the SLMs. **c** Phase-shift scanning in the focal plane of L1, where the $u$ plane is realized by varying the spatial frequency $f_u$. $\theta$ is the diffraction angle between the 0th- and -1st-order diffraction patterns. BS beam splitter, SLM spatial light modulator, sCMOS scientific complementary metal oxide semiconductor.

MCoSM (and other multipoint scanning microscopes) can achieve higher speed and higher SNR by parallelizing the multiple foci excitation (i.e., multiple spot array illumination). However, this increased acquisition speed has some limitations, as the number, resolution, and efficiency of spot arrays are mutually constrained in practical applications.

Through 100–2500 s super-resolution spot illumination with different effective FoVs for imaging, we demonstrate the adjustable capacity of the MCoSM system, as shown in Fig. 2, with an NA of 0.9 at an incident wavelength of 488 nm. The resolutions of the spot arrays are 0.52 Airy, 0.67 Airy, 0.71 Airy, 0.69 Airy, and 0.68 Airy, corresponding to Fig. 2a–e, respectively. The light efficiencies of a $10 \times 10$ array with a resolution of 0.52 Airy and a $50 \times 50$ array with a resolution of 0.68 Airy are 29% and 36%, respectively. Figure 2 shows that the intensity variation differs between the ideal distribution and experimental results; however, the period of the spot array remains unchanged. The PSF of the illumination $p_{\mathrm{illu}}(u,v)$ can be determined without affecting the reconstruction of the super-resolution image, although the generated spot array deviates from the ideal distribution.

The phase-shift scanning method depends on the tilted phase added to the SLM, which is used to replace the transverse mechanical-scan process. To scan the spot array along the $u$ or $v$ plane in the focal plane of the objective, the spatial frequency of the tilted phase term $f_u$, which controls the separation distance (or angle) between the −1st and

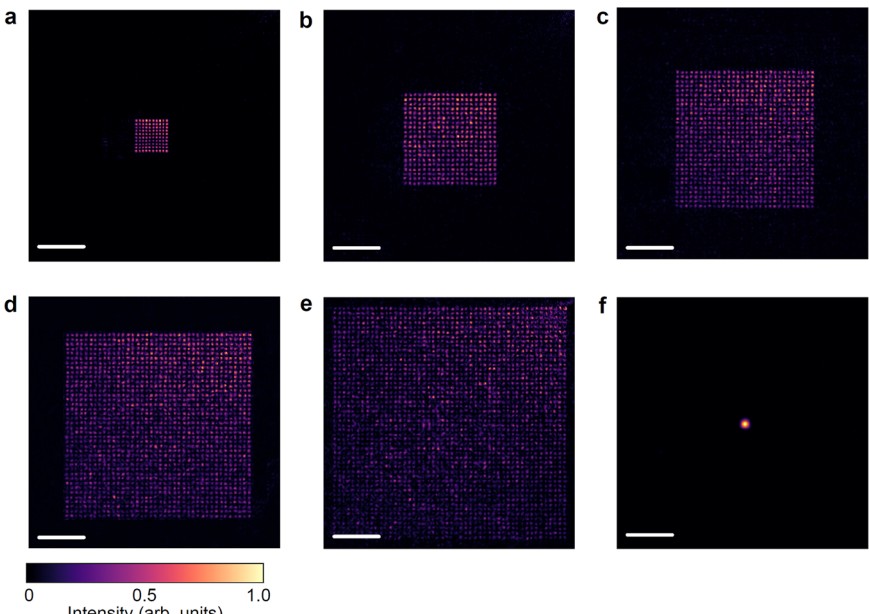

**Fig. 2 | Experimental results of the *N×N* super-resolution spot array.** The results of (**a**) 10 × 10 (0.52 Airy), (**b**) 20 × 20 (0.67 Airy), (**c**) 30 × 30 (0.71 Airy), (**d**) 40 × 40 (0.69 Airy), and (**e**) 50 × 50 (0.68 Airy) arrays and (**f**) the Airy spots with an incident wavelength of 488 nm at an NA of 0.90. Image size: 1500 × 1500 pixels. Scale bar: 1 μm.

0th diffraction orders, can be varied to determine the path in the phase-shift scanning process (Fig. 1b). The $u$ and $v$ planes in the focal plane are independent of the phase-shift scanning process, and the sizes of the spot arrays should be smaller than the maximum scan range[36,39,40]. Small or high spatial frequency values should be avoided, as they may cause the diffracted beams to overlap in space or result in no modulation of the SLM[23]. Therefore, the minimum step size and maximum scan range are of great importance to determine the effective FoV and resolution of the MCoSM system (Fig. 1c). The minimum step size in the phase-shift scanning process $\delta_{u,\min}$ can be expressed as

$$\delta_{u,\min} = \frac{\lambda f_{L1}}{M_{sys}} f_{u,\min} = \frac{\lambda f_{L1}}{M_{sys} L_{u,\max} \Delta p}, \qquad (2)$$

where $f_{u,\min}$ is the minimum frequency of the tilted phase term, $f_{L1}$ is the focal length of lens L1 (Fig. S10), $\Delta p$ is the pixel size of the SLM, $\lambda$ is the wavelength of the incident beam, and $M_{sys}$ is the magnification of the system. The minimum step size is $\delta_{u,\min} = 79.1$ nm when $L_{u,\max} = 1920$ and $\lambda_1 = 405$ nm.

The maximum spatial frequency variation in the tilted phase term can be expressed as $f_{u,\max} = 2/(L_{u,\min} \Delta p)$, and the maximum scan range in the phase-shift scanning process can be expressed as

$$\delta_{u,\max} = \frac{\lambda f_{L1}}{M_{sys}} f_{u,\max} = \frac{2\lambda f_{L1}}{M_{sys} L_{u,\min} \Delta p}. \qquad (3)$$

The maximum scan range ($\delta_{u,\max}$) can reach 210.4 μm, which is slightly smaller than the size of the diffraction field (210.7 μm), by setting $L_{u,\min} = 2$ and $\lambda_4 = 561$ nm. These results indicate that the MCoSM system has suitable scanning resolution for performing quasi-continuous phase shifts. Similarly, the expressions for the minimum step size ($\delta_{v,\min}$) and maximum scan range ($\delta_{v,\max}$) in the $v$ plane are the same as in Eqs. (2) and (3). Notably, aberrations can affect the spot array at the edges of the maximum scan range, especially for large NAs, which practically constrains the scan range (see Supplementary Information 2).

## Fluorescent bead imaging: four-color imaging with 100 nm resolution

To evaluate the spatial and spectral characteristics of the MCoSM system, we performed four-color imaging using fluorescent beads with diameters of 100 nm. The beads were labeled with blue, green, red, or cyan fluorescent dye corresponding to different excitation wavelengths.

In this experiment, the theoretical diffraction-limited resolution was determined by the NA of the objective lens ($NA = 1.25$) and emission wavelength. Here, a 10 × 10 spot array with a resolution of 0.52 Airy was used as the illumination to scan the fluorescent beads (see Visualization 1). The imaging outputs qualitatively indicated that the resolution was higher than the resolution of widefield imaging of the beads (Fig. 3). According to the statistical analysis of the quantitative data derived from 15 single beads, the average size of a single bead in the MCoSM image was reduced to 96 ± 9 nm compared to the size of 197 ± 22 nm in the widefield image at 405 nm. The 197 nm size of the beads in the widefield image was consistent with the 198 nm theoretical diffraction-limited resolution, and the much narrower 96 nm size in the MCoSM image proved that the super-resolution imaging performance can be improved with an enhanced NA. The line profile results indicated that adjacent fluorescent beads could be resolved in the four-color MCoSM results and that super-resolution imaging can be achieved (Fig. 3b–e).

## BPAE imaging: Effective FoV up to 167.4 μm × 167.4 μm with 0.52 Airy resolution

To demonstrate the spatial and spectral characteristics of the MCoSM system with biological samples, we imaged a three-color stained slide of Bovine Pulmonary Artery Endothelial (BPAE) cells, with the nucleus, cytoskeleton, and mitochondria labeled, using our home-built MCoSM prototype. Uploading six sets of phase distributions to the SLMs, we reconstructed super-resolution images illuminated by 10 × 10 ($N = 10$) and 50 × 50 ($N = 50$) spot arrays with resolutions of 0.52 Airy and 0.68 Airy, obtaining 6 images with 2 FoVs with three incident wavelengths ($\lambda_1 = 405$ nm, $\lambda_2 = 488$ nm, and $\lambda_3 = 561$ nm). The captured images (shown in Visualizations 2 and 3) were used as representative examples to illustrate the working principle of the adjustable imaging system. The combined phase distributions were uploaded to the SLMs to scan

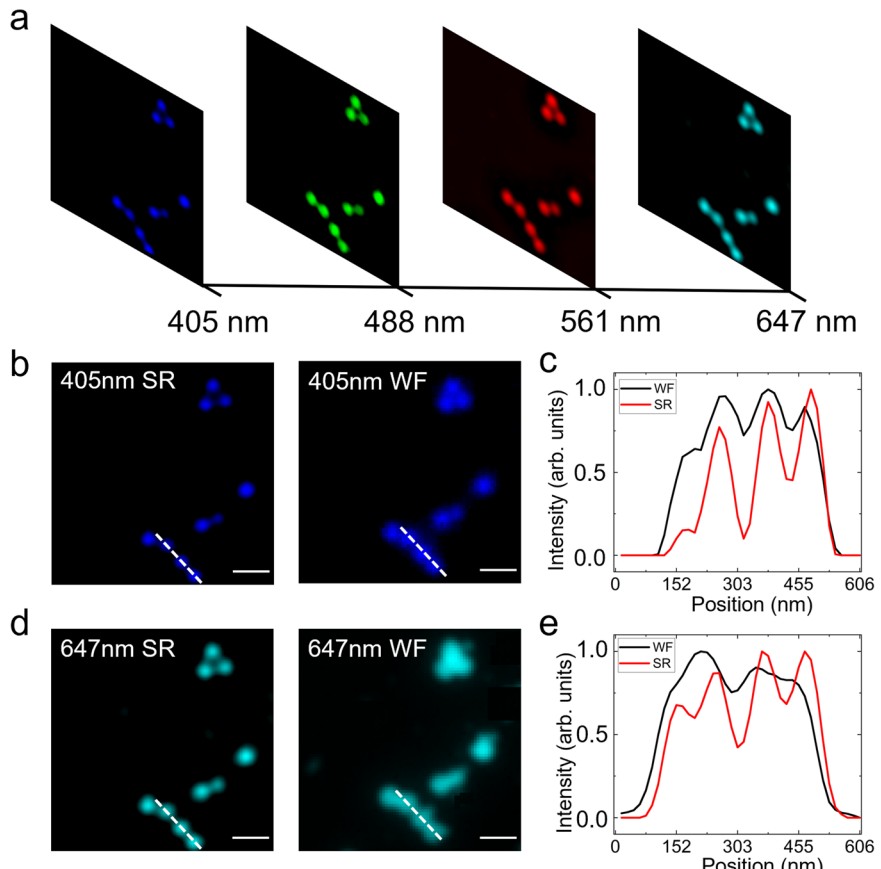

**Fig. 3 | Imaging results of fluorescent beads. a** Four-color fluorescent beads at illumination wavelengths of 405 nm, 488 nm, 561 nm, and 647 nm. **b** and **d** Pseudocolor MCoSM image and widefield image at 405 nm and 647 nm, as marked by the white dotted line. **c** and **e** Intensity profiles of the selected regions shown in (**b**) and (**d**). Scale bar: 10 μm (see Supplementary Movie 1).

the sample, and multicolor super-resolution images were reconstructed based on the spot arrays with a phase-shift scanning method. The nucleus, cytoskeleton, and mitochondria were distinguished in the 2 FoVs images, presenting distinct fluorescence emission properties in the spectral dimension and varied trends (Fig. 4a–c; single wavelength reconstructed images with 10 × 10 spot arrays, incident wavelengths in Fig. S9 of Supplementary Information 3). The distributions of the nucleus, cytoskeleton, and mitochondria over the FoV could be visualized using the reconstructed images obtained at wavelengths of 461 nm (Fig. S9a), 512 nm (Fig. S9b), and 599 nm (Fig. S9c), and the images were created using a mosaic stitching algorithm. The three-color signals were well separated with negligible cross talk.

The MCoSM technique extends the spectral dimension while retaining spatial super-resolution characteristics. Specifically, we extracted the sample curves in the same FoV of the sample, showing that the adjacent cytoskeleton can be successfully resolved in the MCoSM images, which could be mistaken as a single thick strand in the widefield imaging results (Fig. 4a–d). The Gaussian fitting result of the MCoSM intensity profile (Fig. 4d) shows that a lateral resolution of at least 193 nm could be achieved using the MCoSM system with a wavelength of 488 nm, while the corresponding theoretical diffraction-limited resolution was ~347 nm.

Nonetheless, the improved FoV with the proposed MCoSM technique has some limitations. The MCoSM system essentially transforms the majority of the data, which are originally acquired in a mechanical point-by-point manner, into phase-shift scans with the SLM. Considering the pixel size and stability of the SLM, there is a tradeoff between the effective FoV and sampling interval. The effective FoV and sampling interval are determined by the minimum step size (Eq. 2) and

maximum scan range (Eq. 3). To achieve reconstructed images with high SNRs, there is some redundancy during data acquisition, especially for the 10 × 10 super-resolution spot array illumination in the experiments. Hence, the effective FoVs of the multicolor super-resolution images are 61.4 μm × 61.4 μm (Fig. 4a) and 167.4 μm × 167.4 μm (Fig. 4b) for 10 × 10 and 50 × 50 spot arrays with 0.52 Airy and 0.68 Airy resolutions. The effective FoVs are smaller than the linear FoV (500 μm) of the widefield microscope.

## Discussion

The MCoSM system enables multicolor super-resolution microscopy with adjustable resolution and a good effective FoV without mechanical scanning for the first time. The MCoSM system achieves spectral mechanical-scan-free imaging with a time-sharing process involving different color illumination with phase-shift scanning while retaining the spatial characteristics of super-resolution imaging with diffractive spot array illumination. We demonstrated the four-color imaging performance of the MCoSM system using fluorescent beads as a proof of concept. Demonstrations with reference biological samples containing three colors show that the MCoSM system has high resolution and potential in practical applications.

Although the concept of MCoSM was validated in this work, the scan range, speed, spatial characteristics and spectral sampling characteristics can still be improved. First, the maximum scan range and minimum step size are limited by the effective aperture of the SLM. Second, as the phase-shift scanning process can use arbitrary paths in space, typical scanning strategies, e.g., raster, spiral, and Lissajous scanning trajectories, can be reasonably implemented in the SLM-based MCoSM system. Moreover, theoretically, only the adjacent

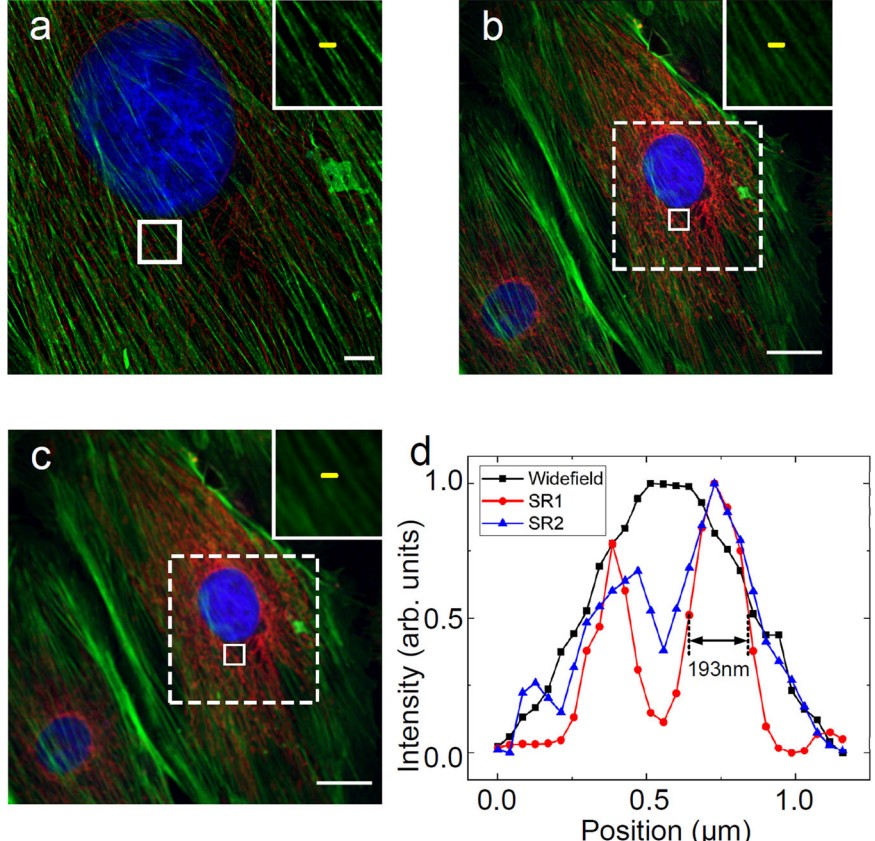

**Fig. 4 | Spatial and spectral characteristics of the MCoSM image of the BPAE.** The combination of reconstructed MCoSM images obtained at wavelengths of 461 nm, 512 nm, and 599 nm, showing the distribution of the nucleus (blue), cytoskeleton (green), and mitochondria (red) over the FoV. **a, b** MCoSM images of organelles obtained by 10 × 10 and 50 × 50 super-resolution spot arrays with resolutions of 0.52 Airy and 0.68 Airy, respectively. **c** Widefield image of organelles obtained with the same objective (objective lens: 60×/0.90 NA). The region of interest (ROI) marked by the white solid rectangle is shown in the enlarged upper right box. Comparisons of identical areas are marked with white dotted rectangles. **d** Intensity profiles of the cytoskeleton on the selected line in (**a**) (SR1), (**b**) (SR2), and (**c**). The effective FoV and spatial resolution of the white dotted box regions in Fig. 3b, c were compared with those of the region in Fig. 3a. Scale bar: (**a**) 5 μm, (**b**) and (**c**) 25 μm (see Supplementary Movies 2 and 3).

region in the spot array must be scanned to satisfy Nyquist sampling; hence, the scanning method can be improved to increase the imaging speed. In addition, the dwell time at each position in the spot array along the scanning path can be controlled, which can enable selective optical stimulation for biomedical samples. By increasing the phase-shift scanning frequency from 10 Hz to 60 Hz, the maximum frame rate of the SLM could be increased, leading to faster imaging and reduced photobleaching. Third, the spatial characteristics could be improved by modifying the optimization algorithm, i.e., higher spatial resolution and more numerous and uniform spots could be achieved to obtain better performance in practice. Although many applications require both a smaller spot size and higher efficiency, there is a tradeoff between spot size and efficiency. Spot arrays with higher spatial resolution and increased uniformity can be generated; however, the FoV should be reduced[24]. Finally, demand for miniaturized super-resolution imaging systems has emerged in recent years, implying diffractive element-based optical devices could be useful across a range of applications[41]. In this context, performance of the MCoSM system could also be further improved with the development of better metasurfaces[25], as metasurfaces could be exploited to further miniaturize the optical setup.

Overall, our findings indicate that multicolor super-resolution microscopy can address prior challenges in the field and be practically implemented with various fluorescent, nanoscale biomedical imaging techniques. Notably, an adjustable illumination strategy may provide a new research direction for performing complex light beam shaping to extend other nanoscopy methodologies, such as multifocal structured illumination microscopy[42,43] and super-resolution scanning microscopy[44–46].

## Methods

### MCoSM setup and image acquisition

Imaging was performed with a lab-built multicolor super-resolution microscope by modifying the Nikon Eclipse Ti inverted microscope with a Nikon perfect focus system (PFS). The excitation sources were matched with emission wavelengths of $\lambda_1 = 405$ nm, $\lambda_2 = 488$ nm, $\lambda_3 = 561$ nm, and $\lambda_4 = 647$ nm (OBIS 405, Sapphire 488, Sapphire 561, OBIS 647; Coherent). A spatial filter consisting of a lens with a focal length of 100 mm and a 5 μm pinhole was used (M-900, Newport). To maintain telecentricity, all distances between lenses were equal to the sum of their respective focal lengths[47]. The three synchronous output beams were combined using three beam splitter cubes (CCM1-PBS251/M, Thorlabs) and temporally synchronized using three spatial light modulators (SLM; PLUTO, Holoeye), with an output energy of ~50 mW.

Different sets of phase distributions (corresponding to different wavelengths) were uploaded to the SLMs to realize super-resolution and phase-shift scanning in the focal plane, where the pixel size and pixel number were 8 μm × 8 μm and 1080 × 1920, respectively. To ensure the same resolution (minimum step and maximum range) in the $u$ and $v$ planes, a circular area with a diameter of 1080 pixels was used. The beam modulated by the SLM was expanded by a telescope to

match the area of the entrance pupil of the illuminating objective. Two lenses were placed in a $4f$ configuration to image the phase distribution at the objective pupil based on the entrance pupil of the microscope objective, where $f_{L1} = 150$ mm and $f_{L2} = 125$ mm, and the zeroth order of the beam was not included in the illumination path with a mounted iris (ID8, Thorlabs). Notably, the amount of phase-shift scanning with multicolor illumination was determined by the phase-shift scanning parameters and incident wavelength.

In the MCoSM system, the same objective, excitation wavelength, and fluorescent filter were used for both in the widefield and reconstructed images (Fig. S10). Removable mirrors (RMs) 1 and 2 (BB05-E02, Thorlabs) were assisted by a removable bracket using an indexing mount (NX1N/M, Thorlabs), which was used to determine the position of the sample using widefield illumination. The high-pass filter was coated by Cr with a thickness of 100 mm and a transmittance of OD3 (0.1%). Fluorescence data were collected through an objective lens with an *NA* of 0.90 and magnification of 60x (RMS60X-PFC, Olympus) and a relay system and recorded by a 2048 × 2048 pixel sCMOS camera (Zyla 4.2 plus, Andor). The system magnification $M_{sys} = 50x$ was determined by the magnification of the objective and $4f$ system. The maximum size of the diffraction field was $\lambda f / \Delta p = 210.7$ μm, which is smaller than the linear FoV of the microscope $d_{field} = \text{FN}/M_{sys} = 500$ μm, with a field number FN = 25 mm, $\lambda = 561$ nm, $f = 3$ mm, and $\Delta p = 8$ μm. The adjustable spot array was realized in a mechanical-scan-free manner, with capture rates of up to 10 frames/s, to ensure delayed reactions and reduced photobleaching of the sample. To adjust the pulse power and time delay to control the SLMs during the phase-shift scanning process, we controlled the MCoSM with Micro-Manager software. In addition, we modulated the SLMs to realize phase switching via XnView software, with the same frame and capture rates. Although the data were somewhat oversampled, we chose the above parameter to meet the Nyquist–Shannon sampling limit of 85 nm and reduce the acquisition time to ensure good imaging performance. The scanning process with a high NA, termed PFS, effectively ensured that the fluorescent sample could be imaged in the experiments with a 167.4 μm × 167.4 μm effective FoV.

The prepared Lumisphere monodisperse microsphere solution consisted of fluorescent beads (7-3-0010, Basel). One milliliter of fluorescent bead solution was dropped on a 0.17 mm cover glass, and the glass was washed 4 times with alcohol buffer before imaging. The diameter of the fluorescent beads was 100 nm, and the beads had blue (350Ex/440Em), green (505Ex/515Em), red (580Ex/605Em), or cyan (633Ex/660Em) labels. We demonstrated the four-color imaging capability of the system, with an objective lens with an NA of 1.25 and a magnification of 60 x (UPLFLN60XOI, Olympus), showing the improved spatial and spectral performance.

We also studied morphologically and ultrastructurally distinct cells of bovine pulmonary artery endothelial (BPAE) slides (F36924, Invitrogen Thermo Fisher) as a practical application of the proposed MCoSM system. Nucleus (Cell-impermeant DAPI, 358Ex/461Em), cytoskeleton (Alexa Fluor 488 phalloidin, 505Ex/512Em), and mitochondria (MitoTracker Red CMXRos, 579Ex/599Em) in the BPAE cells were imaged through an objective lens with an NA of 0.90 and magnification of 60x (RMS60X-PFC, Olympus).

### Phase distribution design

The design of the phase distribution at the SLM is particularly important in the MCoSM system to achieve super-resolution spot array illumination and phase-shift scanning. We used a modified iterative Fourier transform algorithm (IFTA)[48] to develop the super-resolution spot array and introduced a diffractive phase shift grating to achieve mechanical-scan-free imaging.

**Generation of adjustable diffractive spot arrays**. Increasing the number of illumination spots is desirable for improving the effective

FoV; however, achieving this capability in a super-resolution spot array while maintaining good uniformity is challenging[49]. Linear programming has been used to obtain globally optimal phase distributions of SLMs for linearly polarized incident beams with single super-resolution spot[18,28]. However, diffractive optical elements have rarely been used to achieve multi-spot resolution beyond the diffraction limit. Two super-resolution spots were designed by modifying the IFTA algorithm[29]. Notably, Ogura et al. modified the IFTA to design super-resolution spot arrays using DOEs. However, a light efficiency of only ~10% was achieved with a 3 × 3 spot array, with the spot size reduced to 0.8 Airy[50]. Furthermore, our group previously proposed a modified algorithm and experimentally achieved ~30% light efficiency with a 3 × 3 spot array with a spot size of 0.5 Airy, which is undesirable in nanoscale imaging applications[31].

To obtain a super-resolution spot, the sampling interval in the focal plane must be reduced by applying a zero-padding operation to the input plane[31]. The size of the input plane zero-padding was at least 1/8 the size of the original spot to ensure that the spot details could be precisely described. The size of the phase hologram is approximately by $8M_x \times 8M_y$ according to the resampling technique, i.e., the size of the $50 \times 50$ ($N = 50$) spot array is two orders of magnitude larger than the $3 \times 3$ ($N = 3$) spot array if the sampling precision is the same. Hence, a larger spot array exponentially increases the computational overhead, leading to poor spot uniformity, thereby limiting the applicability of such arrays in MCoSM systems. Here, we modified the algorithm by estimating the initial phase and adding an amplitude constraint in the focal plane, and the working principle of the proposed iterative algorithm (Fig. 5a).

We assume that the resolution of the phase distribution is $M_x \times M_y$, the pixel size is $\Delta p$, and the size of the phase hologram is $a \times b$, where $a = M_x \Delta p$ and $b = M_y \Delta p$. During the phase hologram optimization, the pixel size in the hologram plane remains unchanged, while the size in the hologram plane is zero-padded to $8a \times 8b$ to reduce the sampling interval in the focal plane. The number of sampling points in the hologram plane is changed to $8M_x \times 8M_y$, which is the same as the number of sampling points in the focal plane.

The typical choice for the initial complex amplitude in the focal plane is $A_0(x,y) = T_0(x,y) \exp[j\varphi_{spot}^{(0)}(x,y)]$, with an initial amplitude distribution of $T_0(x,y)$ and a random phase of $\varphi_{spot}^{(0)}(x,y)$, where $j = \sqrt{-1}$. However, a random initial phase introduces many phase singularities, which may lead to stagnation problems with the iterative algorithm[51]. To prevent stagnation while ensuring rapid convergence, a two-dimensional estimated quadratic phase was used as the initial phase in the sample plane $(u,v)$ to approximately satisfy these conditions, which can be expressed as

$$\varphi_{spot}^{(0)}(x,y) = C(x^2 + y^2),\qquad(4)$$

where $C$ is a positive coefficient and $x$ and $y$ are the coordinates in the focal plane.

In addition, the weighted constraint strategy is introduced into the iterative process to further enhance the results[52]. The focal plane is partitioned into two regions according to the desired intensity. The areas in the focal plane are divided into two regions (Area I and Area II), and each region is independent (Fig. S1b). The signal region (Area I) is the area where the desired pattern is located, and the non-signal region (Area II) is the area with no desired signal. During each iteration, the enforced amplitude constraint in the focal plane for the next iteration is

$$|A_i(x,y)| = \begin{cases} T_i(x,y), (x,y) \in S \\ \left|A'_{i-1}(x,y)\right|, (x,y) \notin S, \end{cases}\qquad(5)$$

where $S$ denotes the signal region, and the amplitude field $T_i(x,y)$ is replaced in each iteration, adopting negative feedback to improve the

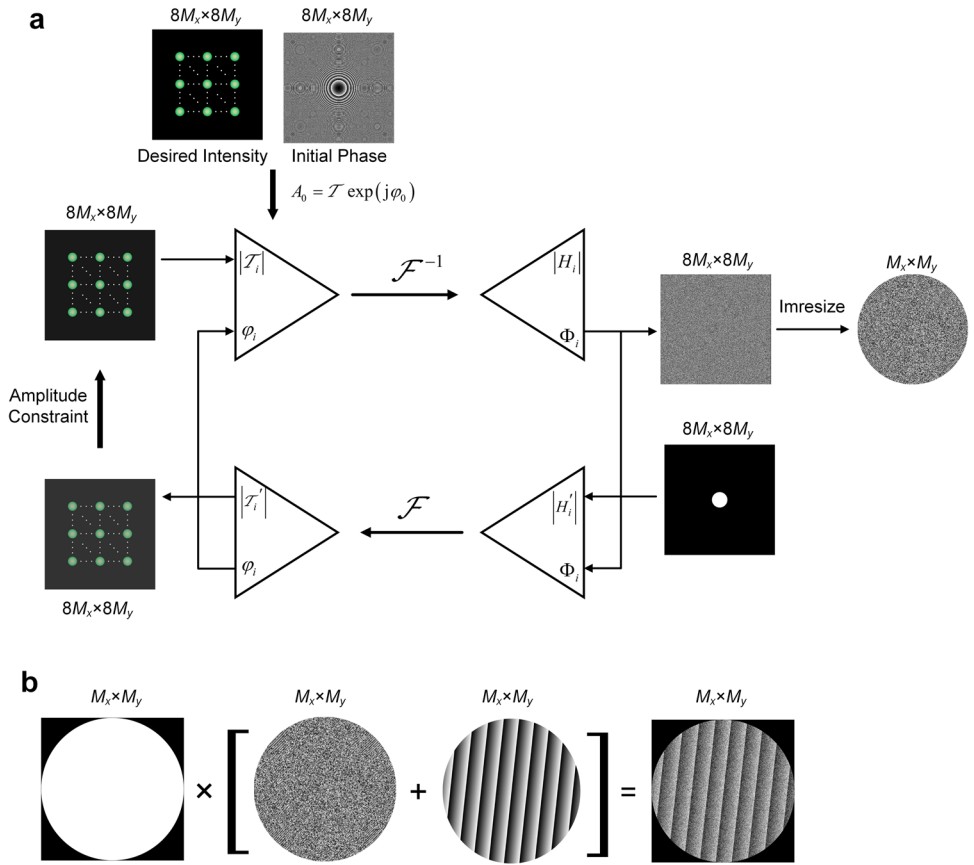

**Fig. 5 | Process flow to produce an $N \times N$ super-resolution spot array with phase-shift scanning.** **a** Flowchart of the iterative algorithm for calculating the phase distribution of a hologram to generate an $N \times N$ super-resolution spot array. **b** Construction of hologram with spot array and phase-shift scanning.

uniformity of the spot array, by the target amplitude as

$$T_i(x,y) = \left[ \sum_n \sqrt{\frac{\langle I_{i-1}(x,y)\rangle_N}{I_{i-1}(x_n,y_n)}} \right] \times T_{i-1}(x,y), \quad (6)$$

where $\langle I_{i-1}(x,y)\rangle_N \approx (1/N)\sum_{n=1}^{N} I_{i-1}(x_n,y_n)$, $T_0(x,y)$ is the desired amplitude distribution, and the phase distribution remains unchanged. In the $i$th iteration, the complex amplitude field $H_i = |H_i| \exp(j\Phi_i)$ in the input plane is calculated by the inverse Fourier transform of the complex amplitude field $A_i$ in the sample plane, and the amplitude distribution is replaced with a uniform amplitude after zero-padding[53]. This iterative process continues until the generated intensity profile in the focal plane converges and satisfies the constraints. The phase hologram can be generated based on the complex amplitude field $H_i$ after phase extraction and cropping operations are performed. The amplitude constraint strategy applied in the focal plane improves the performance of the spot array, leading to faster convergence, by relaxing the amplitude constraint in the non-signal region (in Supplementary Information 4). The convergence of the proposed method is fast and stable, the root mean square error (RMSE) and nonuniformity of our method significantly exceed those of the general IFTA. Moreover, the uniformity of the spot array is effectively improved.

**Generation of phase-shift scanning.** Instead of the typical mechanical scan approach, phase-shift scanning was adopted to realize transverse scanning of the spot array in the focal plane of the objective. According to the diffraction equation, the relationship between the diffractive

angle and wavelength can be expressed as[37]

$$2\Delta p \cdot \sin\Delta\theta = m\Delta\lambda, \quad (7)$$

where $m = 1$, $\theta$ is the obliquity of the added linear phase ramp, and $\Delta p$ is the pixel size of the SLM. The distance between the center of the spot array and the focus of the Fourier lens is given by $\Delta l = f \cdot \tan\Delta\theta$, where $f$ is the focal length of the Fourier lens. Combining the distance with Eq. (7), the designed phase-shift scanning process can be described as

$$\varphi_{\text{ps}}(x,y) = [x \sin(\alpha) + y \cos(\alpha)] \tan\left[ \arcsin\left(\frac{\Delta\lambda}{2\Delta p}\right) \right], \quad (8)$$

where $\alpha = \arcsin(x / \sqrt{x^2 + y^2})$.

**Phase distribution combination.** After combining the above mentioned phase distributions, we added the phases of the spot array $\varphi_{\text{spot}}(x,y)$ and phase-shift scans $\varphi_{\text{ps}}(x,y)$ to the phase distribution uploaded to the phase-only SLM. The phase uploaded to the SLM is described as

$$\varphi_{\text{SLM}}(x,y) = \varphi_{\text{spot}}(x,y) + \varphi_{\text{ps}}(x,y). \quad (9)$$

The complete phase distribution is displayed on the SLM (Fig. 5b). In addition, the experimental results of $10 \times 10$ super-resolution spots with or without a phase shift were analyzed (Fig. S7b, c). There was no significant difference in spot shape or intensity before and after the phase shift was uploaded (see Supplementary Information 2).

## Data processing

**Image pre-processing.** Image pre-processing was divided into five steps. The obtained images were first batch cropped to remove phase-shift scanning edge artifacts. Second, flat-field corrections were applied to the obtained images prior to stitching to correct for illumination spot array inhomogeneity in the same effective FoV. Third, a Gaussian kernel filter was applied for each color to suppress stray light. Fourth, shake correction was applied to address dithering artifacts in the liquid crystal of the SLM and environmental vibrations, which may cause phase shift inaccuracies. Fifth, for raster scanning imaging, the step size of the phase shift was theoretically the same for each row of pixels, and dithering was corrected by averaging the step sizes of adjacent rows.

**Super-resolution image reconstruction.** To reconstruct the sample based on the recorded intensity patterns, we correlated the scanning data with the registration data[54]. We eliminated drifts over long acquisitions via preprocessing techniques. We first calculated the average value of every 200 shots to generate a new set of images. Then, the relative position of each spot was determined to calculate the spot center position, and each spot was registered during the scanning process. By calculating the intensity of the spot centers, the scanned images at each spot can be obtained. Finally, the scanned images of all spots were merged to create a whole image. Notably, the required width of each spot is $S = 1.22\alpha \cdot \lambda \cdot M_{sys}/\text{NA}$, where NA is the numerical aperture of the objective, $\lambda$ is the wavelength of the incident beam, $0 < \alpha < 1$ is the ratio of the spot resolution to the Airy spot resolution, and $M_{sys}$ is the magnification of the system.

**Multicolor stitching.** The excitation and detection efficiency usually depend on the position in the same effective FoV and may differ in multicolor imaging. To obtain multicolor images, the intensity should be corrected in colorimetric resolution analysis. Image analysis was performed using ImageJ (US, National Institutes of Health), Fiji (http://fiji.sc/), and MATLAB (MathWorks).

Multicolor stitching was performed using MATLAB and Fiji after color-balance correction[55]. The lateral position between different color images was evaluated using linear maximum-intensity projections to obtain the optimum multicolor contrast[56,57], and the interactions between images with different fluorescence labels were ignored. The reconstructed images were then divided by the normalized intensity profiles and merged into RGB composites.

## Statistics and reproducibility

Our team was repeated the experiments >20 times of the super-resolution spot arrays illumination in Fig. 2 and imaging results of fluorescent beads in Fig. 3 independently with similar results.

## Reporting summary

Further information on research design is available in the Nature Portfolio Reporting Summary linked to this article.

## Data availability

The data that support the findings of this study are available from the corresponding author upon request.

## Code availability

Custom codes used for analysis and image processing pipelines are available from the corresponding author upon request.

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

## Acknowledgements

We thank Dr. Guoxuan Liu at Huawei Beijing R&D Centre for his helpful assistance during established the experiment and data processing. We also thank Dr. Calum Williams and Dr. Graham Spicer at the Cavendish Laboratory, University of Cambridge, UK for their helpful suggestions during manuscript preparation. This work was supported by the National Natural Science Foundation of China (Grants No. 62075112).

## Author contributions

N.X. wrote the manuscript, conducted theoretical analysis and involved in MCoSM design and imaging experiments. S.E.B. considerably improved the manuscript presentation and the MCoSM imaging mechanism. Z.L. and N.X. contributed to prove the MCoSM mathematically. C.Z. conducted imaging simulations and improved the manuscript. Q.T. initiated the MCoSM imaging idea and supervised the whole project. All authors contributed to the scientific discussion and revision of the article.

## Competing interests

The authors declare no competing interests.
