## [Peer Review File · Nature Communications]

Mechanical-scan-free multicolor super-resolution imaging with diffractive spot array illuminationREVIEWER COMMENTS

Reviewer #1 (Remarks to the Author):

In this manuscript, N. Xu et. al. experimentally proposed a multi-color super-resolution imaging with diffractive spot array illumination based on mechanical-scan-free process, which enables super-resolution imaging with adjustable resolution and field of view. They demonstrated the use of MCoSM for unmixing imaging from four-colors using fluorescent beads as a proof of concept retaining high resolution. In general, the paper is well organized and scientifically sound. Thus, this work is suitable for Nature Communications by addressing the following concerns.

1) Multiple SLMs are used for modulating each laser source. The system of MCoSM will be more concise if it only uses one SLM to implement such function. If not, what is the limitation?

2) The authors have proved mathematically the feasibility of super-resolution imaging with $N \times N$ spot array illumination, which is an impressive work. It is suggested to add some intuitive explanations or concrete examples to help understanding.

3) The intensity distribution of $N \times N$ spot array is asymmetrical of the center position. Can the authors provide further discussion about the concern?

4) The design results also show that there is a contradiction between improving the resolution of the super-resolution spot array and improving the energy utilization rate and reducing the imaging noise. More discussion can be added.

5) More discussion of the illumination quality and potential limitations for further development would be helpful.

6) As the authors mentioned, metasurfaces can be used in further miniaturized optical super-resolution microscopy. Relevant descriptions can be added.

Reviewer #2 (Remarks to the Author):

In the manuscript "Mechanical-scan-free and multi-color super-resolution imaging with diffractive spot array illumination", the authors present an imaging system for multi-color scan-free super-resolution imaging based on a diffractive spot array. The authors overcome

an important challenge in spot or multi-spot excitation schemes, by eliminating the need for scanning with an SLM-induced phase shift. The system relies separate color excitation arms containing separate SLMs, which can in turn allow the tuning of the spot array size and number. The authors make use of the presented methodology to showcase their super-resolution ability by imaging fluorescent beads and fluorescently labeled BPAE cells marking different organelles. Unfortunately, the manuscript is not well written and the concepts are not clearly presented. The presented results are an improvement upon previous uses of diffractive spot arrays, however the results do not clearly demonstrate the benefits of this development compared to the state of the art or a specific application that clearly benefits from this improvement. I recommend additional work to be done in showcasing the wider interest to the scientific community.

Major comments:

- Overall, the manuscript is very difficult to read and I think better justice could be done to the work presented in the paper by improving the clarity. After reading the paper multiple times, I still have difficulties understanding exactly how the authors achieve what they did and how the technological improvement they present compares to the state of the art.
- In the introduction, the authors do not discuss mechanical scanning using a galvanometer scanning mirrors and scan lenses as an option for scanning a multi-spot array. This does not require movement of the stage itself and can produce almost continuous scanning patterns. Instead they just mention scanning via stage movement.
- Equation 1: it is not clear to me where this comes from, or the variables are not clearly defined. The equation suggests that increasing the number of spots will increase the field number or field of view. While I agree that more spots will require less time to image a fixed field-of-view, I don't agree that having more spots allows one to image a larger field of view.
- In the introduction, the authors highlight the importance of the spot size, shape and uniformity when using multi-spot excitation. However, none of these are characterized for this system. A thorough characterization is required to see how these requirements are met and what are the associated artefacts that could arise. Figure 5, which is not referred to in the main text, shows quite significant variation in the intensity of the spots.
- The idea of 'time sharing' is not clearly explained and I still do not understand how authors implement multi-color imaging. This is described as "automatically alternating simultaneous

multimodal acquisitions" which sounds a bit like an oxymoron. Same with "This strategy provides efficient, simultaneous excitation of several fluorescent labels with phase-shifting. In other words, the spatial resolution and spectral channels of MCoSM can be easily altered by switching the parameters of the illumination spot array to meet the requirements (see Methods)." Is it simultaneous? Words like "alternating" and "altered" implies to me that it is changed sequentially. How are the colors combined?

- The authors refer to 'spectral unmixing' in the abstract, but that is not how I see the bead imaging being performed. There are no 'spectra' measured that are then 'unmixed'. Instead it looks more like different colors are acquired separately.

Minor comments:

- What limits the number of spots?

- "Tuning the power of the pulses and their time delay enables control of the excitation windows in a largely independent manner." - how is this done? With the SLM?

- "Super-resolution information is not directly obtained by the imaging system so must be reconstructed". How? This is not described and the following equation just describes the imaging equation of the system? Is deconvolution used?

- "To clarify the super-resolution information also can be obtained when the number of spots extended to N^2 ($N > 5$), we firstly mathematical demonstrated the relative position and intensity of the spots" - this sentence is very hard to understand.

- Parameters in equation (3) are not explained, such as : f_{L1} , $f_{u,min}$, $L_{u,max}$ and Δp . I'm assuming they are related to the SLM dimensions and pitch. Same for equation (4) - some parameters are not defined.

- How is the phase shift scanning implemented? It sounds like the array has to scan the local area first before moving beyond its area.

- "Uploading six stacks phase distribution on the SLMs, we respectively reconstructed super-resolution images illuminated by 10×10 ($N=10$) and 50×50 ($N=50$) spot arrays with a resolution of 0.52λ and 0.68λ illumination, which contained 6 images in 2 FoVs with three-wavelength incidence ($\lambda_1=405\text{nm}$, $\lambda_2=488\text{nm}$, and $\lambda_3=561\text{nm}$)." it is unclear to me what the stacks are referring to. Is it 3 colors for 2 different Airy illumination patterns? Are the stacks phases on the SLM?

As detailed below, we have revised our manuscript in response to the reviewers' comments. The original referee comments are shown in **blue** color, whereas for ease of communication, our answers are provided in **black** color. The changes that we have made in the manuscript text are highlighted in yellow.

Summary of our Revisions:

We have revised our manuscript according to the reviewers' comments, which will be detailed in our specific responses listed below.

We have included additional results on analyses of the relationship between the illumination system and imaging system. We have also modified the introduction section to add more descriptions about point-scanning super-resolution microscopy. In addition, we have included a numerical demonstration of the convolution analysis of super-resolution spot array illumination and imaging system when $N=10$ or $N=50$. Finally, we have also added more descriptions about the characteristics of the multifocal spots. We have revised the manuscript accordingly and added additional discussions in response to the reviewers' comments.

As a quick summary, the following items have been revised or added, all highlighted in yellow in our manuscript files:

Revised sub-sections:

- Changes to the main text:
 - Abstract
 - Introduction
 - Results
 - Discussion and Conclusion
 - Methods
 - References

New sub-sections added and modified:

- New sub-sections added to the supplement note sections:
 - 1.2 Analysis of the relationship between the illumination system and imaging system
- New sub-sections modified to the supplement note sections:
 - 1.3 MCoSM mathematical demonstration: $N \times N$ super-resolution spot array illumination enables super-resolution imaging

New Figures added:

Supplement note Fig. S1: (a) The transmission microscope model and (b) the spot distribution in the focal plane.

Supplement note Fig. S2: Convolution analysis of super-resolution spot array illumination and imaging system. (a) 10×10 ($N=10$) spot array with 0.5 Airy spot size (image size: 600×600 pixels) and (c) 50×50 ($N=50$) spot arrays with 0.7 Airy spot size (image size: 4500×4500 pixels). (b) Airy spot with an image size of 600×600 pixels. (e) is the convolution of (d) and (b) with an image size of 4500×4500 pixels.

Supplement note Fig. S3: Convolution analysis of single super-resolution spot illumination and imaging system.

Reviewer #1 (Comments to the Author):

Overall comments: *In this manuscript, N. Xu et. al. experimentally proposed a multi-color super-resolution imaging with diffractive spot array illumination based on mechanical-scan-free process, which enables super-resolution imaging with adjustable resolution and field of view. They demonstrated the use of MCoSM for unmixing imaging from four-colors using fluorescent beads as a proof of concept retaining high resolution. In general, the paper is well organized and scientifically sound. Thus, this work is suitable for Nature Communications by addressing the following concerns.*

Reply:

We sincerely thank the reviewer for his/her affirmative comments and positive evaluations.

Detailed comments:

Q1. Multiple SLMs are used for modulating each laser source. The system of MCoSM will be more concise if it only uses one SLM to implement such function. If not, what is the limitation?

Reply:

Thank you for your constructive comment and this is an excellent good question. Unfortunately, it is not currently possible to use a single SLM to achieve multi-color super-resolution imaging due to the characteristic of SLM. In the early stages of the experiment, we attempted to use a single SLM for multicolor imaging. In practice, however, the SLM is sensitive to wavelength (especially when the incident wavelength span exceeds 100 nm). It is necessary to match the *Gamma* modulation curve with the incident wavelength. Therefore, we used multiple SLMs to ensure high quality super-resolution imaging. We will explore the possibility of achromatic diffractive optical element, such as Pancharatnam-Berry (PB) metasurface, to generate super-resolution spot array in the future.

Q2. The authors have proved mathematically the feasibility of super-resolution imaging with $N \times N$ spot array illumination, which is an impressive work. It is suggested to add some intuitive explanations or concrete examples to help understanding.

Reply:

We thank the reviewer for their important comment. Following the referee's comments, we have demonstrated a practical application of a single super-

resolution illumination, 10×10 super-resolution spot array illumination, and 50×50 super-resolution spot array illumination to enable super-resolution imaging in numerical simulations, as illustrated in new section 1.2 and Figs. S2 and S3.

The effective PSF of MCoSM can be expressed as Eq. (S1.1). We assume that the NAs of the imaging system and illumination system are the same (e.g. inverted microscope), the image obtained by the sensor, as shown in Fig. S2c, is the convolution of Figs. S2a and S2b. Figure S2e is the convolution of Figs. S2d and S2b, showing the flexible ability of the MCoSM. The separated spot arrays will be blurred after convolution with the imaging system, but it is clear from Figs. S2c and S2e that the relative positions and intensity values of the spot centers will not be affected by convolution because the distance between the adjacent spot centers is twice the size of the super-resolution spot in our model. The NA of the imaging objective equals that of the illumination objective, and as a consequence the super-resolution information can be obtained by the imaging system.

Fig. S2. Convolution analysis of super-resolution spot array illumination and imaging system. (a) 10×10 ($N=10$) spot array with 0.5 Airy spot size (image size: 600×600 pixels) and **(c)** 50×50 ($N=50$) spot arrays with 0.7 Airy spot size (image size: 4500×4500 pixels). **(b)** Airy spot with an image size of 600×600 pixels. **(e)** is the convolution of **(d)** and **(b)** with an image size of 4500×4500 pixels.

Taking a single super-resolution spot with a resolution of 0.5 Airy as an example (the red solid line in Fig. S3). The single super-resolution spot convolution with the imaging system (the blue dotted line in Fig. S3), the

intensity at the detection point becomes slightly wider (the orange solid line in Fig. S3). Hence, the convolution result shows that the performance of a single super-resolution spot is limited by the diffraction limit of the imaging system, but the central position and intensity have not changed. Therefore, the central position and intensity of the obtained spot has not changed even though the illumination spot at the detection point is restricted by the imaging system after convolution and the obtained spot is not super-resolved in a single super-resolution ($N=1$). Similarly, the obtained intensity of the central spot can also be extracted to obtain the super-resolution information of the sample. It should be noted that Fig. S3 does not consider the influence of the sidelobes of the super-resolution spot and other orders of Airy spots after convolution.

Fig. S3. Convolution analysis of single super-resolution spot illumination and imaging system. Airy spot: 1000×1 pixels, single super-resolution spot: 500×1 pixels.

By reducing the size of the illumination spots matrix and increasing the NA of the imaging system, the overlapping area of the spot array becomes smaller after imaging with separated super-resolution spot array. However, the determination of the improvement of resolution is based on the imaging objective. The best match is the NA of the imaging objective and the illumination objective are the same. Therefore, an inverted microscope shared by the illumination/imaging objective can be used to construct a super-resolution microscopic imaging system, and the super-resolution spot array needs to be sparse.' have been added to the revised manuscript. Please see pages S4-S6, lines 43-88.

We also clarified and modified the '1.3 MCoSM mathematical demonstration: $N \times N$ super-resolution spot array illumination enables super-resolution imaging'-section as:

'In sections 1.1 and 1.2, we have analyzed the relationship between the illumination system and imaging system, and the numerical simulation verifies that the super-resolution imaging can be realized under the conditions of single super-resolution spot ($N=1$), 10×10 ($N=10$), and 50×50 ($N=50$) spot array illumination. We wondered whether the process for $N \times N$ superresolution spot arrays was the same as that for 3×3 or 5×5 spot arrays after the convolution. Moreover, we wondered whether the position and intensity of the center spot (after normalization) remained constant.' Please see page S7, lines 93-99.

Q3. The intensity distribution of $N \times N$ spot array is asymmetrical of the center position. Can the authors provide further discussion about the concern?

Reply:

We sincerely appreciate the precious feedback from the reviewer. We agree with the reviewer that there is a bit of confusion about the delineation of the target field. Additional explanation is provided here. The focal plane is divided into two regions (Area I and Area II), and each region is independent, as shown in Fig. S1b. Area I is the super-resolution spot array, Area II is the background area.

In line with the reviewer's comment, we have added an image about Area division in supplement note Fig. S1b. *"The areas in the focal plane are divided into two regions (Area I and Area II), and each region is independent (Fig. S1b). The signal region (Area I) is the area where the desired pattern is located, and the non-signal region (Area II) is the area with no desired signal."* have been added to the revised manuscript. Please see page 17, lines 8-11.

Following the referee's suggestion, we have also added more descriptions and drawn Fig. S1b in the 'supplement note 1.1'-section as follows:

'Generally, the pixel size of the focal plane is restricted by the equation $\Delta x = \lambda f / N \cdot \Delta p$, where Δp is the pixel size of the SLM, λ is the incident wavelength, f is the focal length of the objective, N is the sampling number of the focal plane, and N and Δp are equal to 2048 and $8 \mu\text{m}$ in the MCoSM, respectively. The distance between the central zeroth order beam and spot array center is $z = 200 \times \Delta x$. The angle of the spot array laterally offset relative to the axes of the objective is $\theta = \arctan(z / f) = \arctan(200\lambda / N \cdot \Delta p)$.

When the incident wavelength λ_1 is equal to 632.8 nm, θ_1 is equal to 0.44° ; when the incident wavelength λ_2 is equal to 405 nm, θ_2 is equal to 0.28° . We believe the lateral offset of the spot array is not affected by the MCoSM, which is consistent with experimental results. The spot array is away from the spatial

zeroth order to ensure the high-quality illumination (Fig. S1b).’ Please see page S3, lines 26-37.

Q4. The design results also show that there is a contradiction between improving the resolution of the super-resolution spot array and improving the energy utilization rate and reducing the imaging noise. More discussion can be added.

Reply:

We thank the reviewer for his/her constructive feedback. Yes, illumination quality is largely constrained between the resolution and efficiency of the optimization algorithm. Since the trade-off between the resolution and efficiency is largely influenced by the designed algorithm, the relationship curve between the resolution and efficiency is limited useful for improving the optimization algorithm.

Following the referee’s suggestion, we have added more descriptions about the development of super-resolution spot array in the ‘Phase distribution design’-section as follows:

‘Linear programming has been used to obtain globally optimal phase distributions of SLMs for linearly polarized incident beams with single super-resolution spot [18, 28]. However, diffractive optical elements have rarely been used to achieve multi-spot resolution beyond the diffraction limit. Two super-resolution spots were designed by modifying the IFTA algorithm [29]. Notably, Ogura et al. modified the IFTA to design super-resolution spot arrays using DOEs. However, a light efficiency of only approximately 10% was achieved with a 3x3 spot array, with the spot size reduced to 0.8 Airy [50]. Furthermore, our group previously proposed a modified algorithm and experimentally achieved ~30% light efficiency with a 3x3 spot array with a spot size of 0.5 Airy, which is undesirable in nanoscale imaging applications [31].’ Please see page 15, lines 2-11.

Q5. More discussion of the illumination quality and potential limitations for further development would be helpful.

Reply:

We sincerely thank the reviewer for the valuable suggestion. The illumination quality affects the imaging, which mainly includes interdependent factors such as resolution, field of view, imaging speed, and illumination intensity. Currently, a 50x50 super-resolution spot array with a resolution of 70% of the Airy size or

a 10×10 super-resolution spot array with a resolution of 50% of the Airy size can be realized by diffractive optical element. If the number and resolution of spots can be increased, it may be possible to obtain higher resolution and larger field of view. However, the number, efficiency, and resolution of super-resolution spot array exists tradeoff. We will improve the algorithm to further increase the resolution and the number of spot array in the future. Following the referee's suggestion, we have added the potential limitation for further development in the 'Discussion and Conclusion'-section as follows:

'Although many applications require both a smaller spot size and higher efficiency, there is a tradeoff between spot size and efficiency. Spot arrays with higher spatial resolution and increased uniformity can be generated; however, the FoV should be reduced [24]. Finally, demand for miniaturized super-resolution imaging systems has emerged in recent years, implying diffractive element-based optical devices could be useful across a range of applications [41].' Please see page 12, lines 17-22.

Q6. As the authors mentioned, metasurfaces can be used in further miniaturized optical super-resolution microscopy. Relevant descriptions can be added.

Reply:

We sincerely thank the reviewer for the valuable suggestion. Following the referee's suggestion, we have added more sentences and references to describe the miniaturized optical super-resolution microscopy in the 'Introduction'-section as such:

'Flexible phase distributions can be used to generate a range of modulated intensity distributions by harnessing the power and affordability of diffractive optics, leading to the development of new illumination strategies for optical microscopy and nanoscopy; moreover, mass production is feasible via nanoprinting, with the potential cost per element reduced to only a few dollars [23, 24]. In particular, metasurfaces can be used to develop miniaturized devices with high resolution [25-27].' Please see page 3, lines 2-7.

We also clarified the 'Discussion and Conclusion'-section as:

'Finally, demand for miniaturized super-resolution imaging systems has emerged in recent years, implying diffractive element-based optical devices could be useful across a range of applications [41]. In this context, performance of the MCoSM system could also be further improved with the development of better metasurfaces [25], as metasurfaces could be exploited to further miniaturize the optical setup.' Please see page 12, lines 20-24.

Reviewer #2 (Comments to the Author):

Overall comments: *In the manuscript 'Mechanical-scan-free and multi-color super-resolution imaging with diffractive spot array illumination', the authors present an imaging system for multi-color scan-free super-resolution imaging based on a diffractive spot array. The authors overcome an important challenge in spot or multi-spot excitation schemes, by eliminating the need for scanning with an SLM-induced phase shift. The system relies separate color excitation arms containing separate SLMs, which can in turn allow the tuning of the spot array size and number. The authors make use of the presented methodology to showcase their super-resolution ability by imaging fluorescent beads and fluorescently labeled BPAE cells marking different organelles. Unfortunately, the manuscript is not well written and the concepts are not clearly presented. The presented results are an improvement upon previous uses of diffractive spot arrays, however the results do not clearly demonstrate the benefits of this development compared to the state of the art or a specific application that clearly benefits from this improvement. I recommend additional work to be done in showcasing the wider interest to the scientific community.*

Reply:

We appreciate the constructive feedback, and we have added detailed analyses to the revised manuscript.

Point-scanning microscopy is indispensable for investigating nanoscale imaging. However, the reported point-scanning super-resolution imaging technology has some shortcomings. Among the various techniques, confocal microscopy can achieve at best a $\sqrt{2}$ times resolution with one spot via a focused beam and pinhole detection. SIM increases the cutoff frequency of the optical transfer function to obtain theoretical double-resolution, achieving widefield imaging at relatively low phototoxicity. The performance of STED microscopy methods scales with the scales with illumination intensity; moreover, although such approaches constrain the choice of dyes and must balance the spatial resolution and phototoxicity, they enable cellular imaging with a resolution as low as 20 nm.

The MCoSM system has a simple structure. By inserting a phase-only diffractive optical element (DOE) into the illumination system of a standard microscope, super-resolution information of the sample can be obtained based on super-resolution spots. The resolution and field of view of the MCoSM system can be adjusted (reducing the number of spots can improve the resolution to 69 nm [1]). Moreover, this technology has no requirements for the fluorescent dyes and can theoretically achieve label-free super-resolution imaging (this benefit is not the focus of this manuscript, and we have reported

on the capacity of a 3×3 super-resolution spot array to realize label-free imaging with increased resolution [2]). In addition, we implement a phase-shift scanning technique instead of a mechanical scanning approach to reduce the damage of mechanical movement to the sample. Our findings indicate that multicolor super-resolution microscopy can address prior challenges encountered in the field and be practically implemented in a variety of fluorescent, nanoscale biomedical imaging applications. Moreover, an adjustable illumination strategy may provide a new research direction for performing complex light beam shaping and extending other nanoscopy methodologies.

In line with the reviewer's suggestion, we have rewritten the beginning of the 'Introduction'-section to clearly highlight the benefit relative to the current state of the art as follows:

'Point-scanning super-resolution microscopy, such as confocal microscopy (CM) [1, 2], structured illumination microscopy (SIM) [3], and stimulated emission depletion (STED) microscopy [4], has become indispensable for investigating nanoscale biological structures and interactions in biomedical research. Among the various techniques, CM physically provides a times resolution at best with one spot via a focused beam and pinhole detection. SIM increases the cutoff frequency of the optical transfer function to obtain theoretical double-resolution, achieving widefield imaging at relatively low phototoxicity [5]. The performance of STED microscopy scales with the illumination intensity; moreover, although such approaches constrain the choice of dyes and must balance the spatial resolution and phototoxicity, which enables cellular imaging with a resolution as low as 20 nm [6].

For multicolor super-resolution imaging, the use of distinct fluorescent labels enables researchers to study spatial relationships between different molecular processes within cells [7]. Traditional point-scanning super-resolution microscopy can be challenging when applied for multicolor imaging due to the long data acquisition time, which normally exacerbates photobleaching. In addition, movement of the microscope stage typically requires mechanical adjustments, which can perturb the sample during refocusing, although this issue can sometimes be mitigated with remote focusing [8, 9]. Notably, galvanometer scanners [10] and scan lenses [11, 12] are commonly used to achieve relative displacements between the sample and illumination system, which inevitably increases in size. These issues have led to difficulty in achieving multicolor fluorescence imaging across wide fields of view with nanometer-scale precision in biological samples.' in the revised manuscript. Please see pages 1 and 2, lines 29-33, 1-17.

We polished the revised manuscript with professional assistance, consulting relevant literature and Springer Nature Gold Language Editors. We have carefully studied your suggestions, and the detailed replies and modifications are presented below. If you have any other questions or ideas to discuss with us, we welcome further discourse following the progress of the article through the review process or beyond!

References

[1] N. Xu, G. Liu, and Q. Tan, "High-fidelity far-field microscopy at $\lambda/8$ resolution," *Laser Photonics Rev.* **16**, 2200307 (2022).

[2] N. Xu, G. Liu, and Q. Tan, "Adjustable super-resolution microscopy with diffractive spot array illumination," *Appl. Phys. Lett.* **116**, 254103 (2020).

Detailed comments:

Q1. Overall, the manuscript is very difficult to read and I think better justice could be done to the work presented in the paper by improving the clarity. After reading the paper multiple times, I still have difficulties understanding exactly how the authors achieve what they did and how the technological improvement they present compares to the state of the art.

Reply:

Following the referee's comment, we respectfully explain here in brief the innovations of our work and how the proposed system can be implemented.

- The innovation of our work is as follows:

(a) We design and build a mechanical-scan-free multicolor super-resolution microscopy (MCoSM) system *for the first time*, which enables mechanical-scan-free superresolution imaging with adjustable resolution and a good effective field of view (FoV) based on spatial light modulators (SLMs).

(b) We imaged fluorescent beads and biological tissues using the MCoSM system, and the maximum scanning range and minimum step size were 210.4 μm and 79.1 nm on both the x and y axes, respectively.

(c) The MCoSM system achieved four-color imaging in experiments, and the resolution was higher than 96 nm using a 10×10 spot array, with a resolution of 52% of the Airy spot (0.52 Airy).

- How is the MCoSM implemented?

(a) Super-resolution imaging: We generate 100-2,500 super-resolution spots for illumination, with different effective FoVs achieved with a diffractive optical

element. To ensure the super-resolution performance, the spot arrays should be resolved in the imaging system. We match the illumination and imaging parameters for resolved spots in the MCoSM system.

(b) Mechanical-scan-free imaging: Lateral scanning of the sample can be realized by adding the tilted phase on the SLMs. See supplement note 2 for the characteristics (frequency, intensity, and profile) of spot array before and after adding tilted phase on the SLMs.

(c) Multicolor imaging: Multicolor imaging is achieved by automatically alternating the multimodal acquisitions in sequence.

Following the referee's suggestion, we have added detailed information to the beginning of the 'Introduction' section and emphasized the limitations that are being overcome with our implementation in the last paragraph. We believe that this now better shows the efficacy of the MCoSM system relative to the current state of the art.

Q2. In the introduction, the authors do not discuss mechanical scanning using a galvanometer scanning mirrors and scan lenses as an option for scanning a multi-spot array. This does not require movement of the stage itself and can produce almost continuous scanning patterns. Instead they just mention scanning via stage movement.

Reply:

We agree with the reviewer's suggestions. Following the referee's suggestion, we have added the galvanometer scanning mirrors and scan lenses techniques in the 'Introduction'-section as follows:

'Notably, galvanometer scanners [10] and scan lenses [11, 12] are commonly used to achieve relative displacements between the sample and illumination system, which inevitably increases in size.' Please see page 2, lines 13-15.

Q3. Equation 1: it is not clear to me where this comes from, or the variables are not clearly defined. The equation suggests that increasing the number of spots will increase the field number or field of view. While I agree that more spots will require less time to image a fixed field-of-view, I don't agree that having more spots allows one to image a larger field of view.

Reply:

We thank the reviewer for his or her constructive feedback. Increasing the number of spot arrays increases the effective field of view (FoV) for multiple

measurements, which results in shorter measurement times.

In general, the FoV of a microscopic system is limited by the pixel size, wavelength, and numerical aperture of the system. In terms of the diffraction field, the FoV of the microscopic system is fixed after the system parameters are determined. The spot array produced by the DOE is the actual sampling FoV (called the effective FoV), and its size is limited by the size of the diffraction field.

We agree with the reviewer's suggestion. To distinguish between the FoV of the microscopic system and FoV of the spot array, we have modified the 'FoV' of the spot array to the '*effective FoV*' in the revised manuscript. However, the effective FoV is still limited by the size of the diffraction field. In addition, we analyze the numerical calculation of the size of the diffraction field and microscopic FoV in our response to Q7.

Following the reviewer's comment, in our revised manuscript, we highlighted in the 'Introduction' section the fact that increasing the number of spot arrays increases the effective FoV while leading to shorter measurement times as follows:

'For a Fourier transform system, the maximum size of the diffraction field without aliasing in the output plane is restricted by $\lambda f / \Delta p$, where f is the focal length of the objective and Δp is the sampling interval of the diffractive optical element (DOE) [14]. Generally, the spot array does not fully use the entire diffractive field, and the effective FoV is determined by the number of spot arrays.' Please see page 2, lines 26-30.

Q4. In the introduction, the authors highlight the importance of the spot size, shape and uniformity when using multi-spot excitation. However, none of these are characterized for this system. A thorough characterization is required to see how these requirements are met and what are the associated artefacts that could arise. Figure 5, which is not referred to in the main text, shows quite significant variation in the intensity of the spots.

Reply:

We agree with the reviewer's suggestions. For a point-scanning microscope, the characteristics of the spots (spot size, number, efficiency, and uniformity) are important when using multispot excitation. The artifacts in the images obtained with different spot array illumination techniques are not the focus of the present manuscript. A quantitative analysis would be difficult, as this process involves nonlinear fluorescence effects. Following the referee's suggestion, we have added a qualitative explanation by rewriting the middle of

the 'Principles of MCoSM'-section as follows:

'The MCoSM system obtains scans of the sample with multiple excitations, and the resulting fluorescence at each scan position is mapped to the corresponding pixels in the image. It is often assumed that the speed of single spot illumination methods ($N=1$ in the MCoSM system) can be improved by increasing the scan speed; however, the resulting decrease in the per-pixel dwell time reduces the quality of the overall signal and degrades the image's signal-to-noise ratio (SNR) [38]. Increasing the illumination intensity compensates for this effect but can lead to increased photodamage and photobleaching (and at high intensities, these processes can scale nonlinearly with intensity). The MCoSM (and other multipoint scanning microscopes) can achieve higher speed and higher SNR by parallelizing the multiple foci excitation (i.e., multiple spot array illumination). However, this increased acquisition speed has some limitations, as the number, resolution, and efficiency of spot arrays are mutually constrained in practical applications.' Please see page 6, lines 2-13.

Regarding Fig. 5, as the reviewer commented, the super-resolution spot array shown in Fig. 5 is crucial to the performance of the MCoSM system, reflecting the resolution and number of spot arrays. Originally, we intended to refer to the 'Methods' section for a verification of the phase design results. We agree with the reviewer's suggestion and have referred to the main text in the 'Principle of MCoSM' section and added descriptions to the revised manuscript.

'Through 100-2,500 s super-resolution spot illumination with different effective FoVs for imaging, we demonstrate the adjustable capacity of the MCoSM system, as shown in Fig. 2, with an NA of 0.9 at an incident wavelength of 488 nm. The resolutions of the spot arrays are 0.52 Airy, 0.67 Airy, 0.71 Airy, 0.69 Airy, and 0.68 Airy, corresponding to Figs. 2a-2e, respectively. The light efficiencies of a 10×10 array with a resolution of 0.52 Airy and a 50×50 array with a resolution of 0.68 Airy are 29% and 36%, respectively. Figure 2 shows that the intensity variation differs between the ideal distribution and experimental results; however, the period of the spot array remains unchanged. The PSF of the illumination $p_{\text{illu}}(u,v)$ can be determined without affecting the reconstruction of the super-resolution image, although the generated spot array deviates from the ideal distribution.' Please see page 6, lines 14-23.

Q5. The idea of 'time sharing' is not clearly explained and I still do not understand how authors implement multi-color imaging. This is described as 'automatically alternating simultaneous multimodal acquisitions' which sounds a bit like an oxymoron. Same with 'This strategy provides efficient, simultaneous excitation of several fluorescent labels with phase-shifting. In

other words, the spatial resolution and spectral channels of MCoSM can be easily altered by switching the parameters of the illumination spot array to meet the requirements (see Methods).’ Is it simultaneous? Words like ‘alternating’ and ‘altered’ implies to me that it is changed sequentially. How are the colors combined?

Reply:

We thank the reviewer for providing the opportunity to better explain the MCoSM system and apologise for the original lack of clarity. Here, we want to express the meaning of the continuous data collection approach based on a time sharing strategy, which as the reviewer notes, means wavelengths are changed sequentially. The acquisition at one wavelength starts immediately after the acquisition at another wavelength. We agree with the reviewer’s suggestions and have modified the relevant statement to be more clear and accurate. In addition, *‘Typically, multicolor images are routinely obtained by employing a time-sharing process to acquire the expected organelle information with different incident wavelengths applied sequentially [32, 33].’* has been added in the revised manuscript. Please see page 3, lines 18-20.

Multicolor stitching was performed using ImageJ Fiji software. The detailed process is described in the Data processing section, part C.

Q6. The authors refer to ‘spectral unmixing’ in the abstract, but that is not how I see the bead imaging being performed. There are no ‘spectra’ measured that are then ‘unmixed’. Instead it looks more like different colors are acquired separately.

Reply:

We thank the reviewer for noting this misleading statement. We agree with the reviewer’s suggestions and have modified the relevant statement to *‘multicolor imaging’*. In the MCoSM system, multiple wavelengths are used to extract different information, and the extraction process does not involve *‘spectral unmixing’*. However, multicolor stitching is required when reconstructing the image of the BPAE sample (Figure 4). The stitching step is achieved using Fiji ImageJ software. We have presented this information in the data processing section.

Following the reviewer’s comment, we have modified in our revised manuscript, *‘To demonstrate the prospects of MCoSM, we perform four-color imaging of fluorescent beads at high resolution.’* and *‘We demonstrated the four-color imaging performance of the MCoSM system using fluorescent beads as a proof of concept. Demonstrations with reference biological samples containing three*

colors show that the MCoSM system has high resolution and potential in practical applications.' Please see pages 1 and 11-12, lines 24-26 and 19-20, 1-2, respectively.

Q7. What limits the number of spots?

Reply:

This is a good question. The number of spots is mainly limited by two aspects, one is the size of the diffraction field, and the other is the limitation of the quality of the spots.

The size of the field of view (FoV) of a microscope can be characterized by the linear FoV. The linear FoV of the Nikon Eclipse Ti microscope can be expressed as $d_{\text{field}} = 25\text{mm}/M_{\text{sys}}$ (widefield, diffraction-limited resolution). The magnification of the MCoSM system, $M_{\text{sys}} = 50\times$, is determined by the magnification of the objective (60x) and the $4f$ system, which is $d_{\text{field}} = 500\ \mu\text{m}$.

The maximum effective FoV of the diffraction field is half of the main lobe of the **sinc** function [1]. The size of the diffraction field is limited by the pixel size of the diffractive optical element (DOE), incident wavelength and focal length of the objective. For DOEs with plane wave incidence, the maximum size of the diffraction field without aliasing in the output plane is restricted by $\lambda f / \Delta p$, where λ is the wavelength of the incident beam, f is the focal length of the objective, and Δp is the sampling interval of the DOE. For the MCoSM system, the maximum size of the diffraction field is $210.7\ \mu\text{m}$ when $\lambda = 561\ \text{nm}$, $f = 3\ \text{mm}$, and $\Delta p = 8\ \mu\text{m}$, which is smaller than $d_{\text{field}} = 500\ \mu\text{m}$. However, the effective FoVs of the multicolor super-resolution images are $61.4\ \mu\text{m} \times 61.4\ \mu\text{m}$ (Fig. 3a) and $167.4\ \mu\text{m} \times 167.4\ \mu\text{m}$ (Fig. 3b). The spot sizes cannot exceed the size of the diffraction field, which is due to the basic limitation of the diffraction field and **the effective FoV, which is determined by the number of spot arrays.**

In addition, the number of spots is limited by the quality of the spot array, and the quality of the spot array is largely affected by the optimization algorithm. The quality of the spot array includes factors such as the resolution, signal-to-noise ratio, and number of spots. In the proposed optimization algorithm, a single super-resolution spot with an Airy spot size resolution of 0.38 can be realized with an efficiency of $\sim 3.76\%$ [2]; moreover, a 3×3 superresolution spot array with an Airy spot size resolution of 0.5 can be realized with an efficiency of $\sim 32\%$ [3]. In the MCoSM system, a 10×10 superresolution spot array with an Airy spot size resolution of 0.52 can be realized with an efficiency of $\sim 29\%$, and a 50×50 superresolution spot array with an Airy spot size resolution of 0.68 can be realized with an efficiency of $\sim 36\%$. If the efficiency requirements can be

relaxed, the number of spots can be increased.

In addition, dark regions and constrained regions are added surrounding the effective diffraction field to achieve a superresolution spot array with sufficient brightness and uniformity. In our opinion, the results are quite good, and it would be difficult to further increase performance (number and efficiency). We will endeavor to improve the performance of the algorithm in future work.

Following the reviewer's comment, we have also added the following clarification in our revised manuscript at the 'Results'-section and 'Methods'-section:

'The light efficiencies of a 10x10 array with a resolution of 0.52 Airy and a 50x50 array with a resolution of 0.68 Airy are 29% and 36%, respectively.' Please see page 6, lines 17-18.

'The maximum scan range ($\delta_{u,\max}$) can reach 210.4 μm , which is slightly smaller than the size of the diffraction field (210.7 μm), by setting $L_{u,\min} = 2$ and $\lambda_4 = 561 \text{ nm}$.' and *'The effective FoVs are smaller than the linear FoV (500 μm) of the widefield microscope.'* Please see pages 8 and 10, lines 9-10 and 28-29, respectively.

'The maximum size of the diffraction field was $\lambda f / \Delta p = 210.7 \mu\text{m}$, which is smaller than the linear FoV of the microscope $d_{\text{field}} = \text{FN}/M_{\text{sys}} = 500 \mu\text{m}$, with a field number $\text{FN}=25 \text{ mm}$, $\lambda = 561 \text{ nm}$, $f = 3 \text{ mm}$, and $\Delta p = 8 \mu\text{m}$.' Please see page 13, lines 32-34.

References

- [1] D. Mas, J. Garcia, C. Ferreira, L. M. Bernardo, and F. Marinho, "Fast algorithms for free-space diffraction patterns calculation," *Opt. Comm.* **164**, 233-245 (1999).
- [2] N. Xu, G. Liu, and Q. Tan, "High-fidelity far-field microscopy at $\lambda/8$ resolution," *Laser Photonics Rev.* **16**, 2200307 (2022).
- [3] N. Xu, G. Liu, and Q. Tan, "Adjustable super-resolution microscopy with diffractive spot array illumination," *Appl. Phys. Lett.* **116**, 254103 (2020).

Q8. 'Tuning the power of the pulses and their time delay enables control of the excitation windows in a largely independent manner.' - how is this done? With the SLM?

Reply:

We thank the reviewer for allowing us to better explain the operation. PLUTO

SLMs have no trigger interface for their supporting software and hardware, and they cannot be directly controlled based on the power of the pulses. However, to tune the pulse power and time delay, which enables control of the SLM with multiple phase scanning processes, we controlled the SLM with XnView software. XnView software can be used to control the timing of phase switching instead of PLUTO software. We used Micro-Manager and XnView software to adjust the power, control the time delay and achieve phase switching.

Following the referee's suggestion, we have added the following paragraph to the 'Methods'-section regarding this point:

'To adjust the pulse power and time delay to control the SLMs during the phase-shift scanning process, we controlled the MCoSM with Micro-Manager software. In addition, we modulated the SLMs to realize phase switching via XnView software, with the same frame and capture rates.' Please see page 14, lines 2-6.

We have also added the following clarification at the beginning of the 'Results'-section to reduce any reader confusion regarding these points:

'The incident wavelength $\lambda_n (n \in \mathbb{N}^)$ can be adjusted to achieve multicolor excitation processes and multicolor phase-shift scanning with the excitation of the corresponding chromophores.'* Please see page 4, lines 18-20.

Q9. 'Super-resolution information is not directly obtained by the imaging system so must be reconstructed'. How? This is not described and the following equation just describes the imaging equation of the system? Is deconvolution used?

Reply:

We thank the reviewer for this important comment. The reconstruction algorithm used in the manuscript does not use a deconvolution algorithm. Instead, the image is directly reconstructed using the superposition of the intensity distribution without involving deconvolution operations.

The 'Data Processing'-section in the manuscript provides a detailed description of the reconstruction process. First, the relative position of each spot is determined to calculate the spot center position, and each spot is registered during the scanning process. By calculating the intensity of the spot centers, the scanned images at each spot can be obtained. Finally, the scanned images of all spots are merged to generate the whole image. The reconstruction method can be summarized as *'determine the center position of the spots and extract the central intensity'*. In other words, we have improved the resolution

by physical means without using deconvolution or other methods to improve the resolution.

Regarding the applicability to the super-resolution reconstruction processing, we have added the following 'Principle of MCoSM'-section paragraph to our revised manuscript:

'The reconstruction process can be summarized as 'determine the center position of the spots and extract the central intensity'. Thus, we improved the resolution by physical means without using deconvolution or other processes (see Data processing).' Please see page 4, lines 24-27.

Q10. 'To clarify the super-resolution information also can be obtained when the number of spots extended to N^2 ($N>5$), we firstly mathematical demonstrated the relative position and intensity of the spots' - this sentence is very hard to understand.

Reply:

We apologise for the lack of clarity. Although we modified the algorithm to obtain super-resolution spot arrays with relatively large effective FoVs and small spot sizes with the chosen illumination beams, the spot arrays should be resolved in the imaging system to ensure the super-resolution performance of the MCoSM system. Hence, to obtain a resolved spot array, we must match the illumination system with the imaging system. We adopt the convolution relationship of the PSF to characterize the mathematical relationship of the MCoSM system, as shown in Eq. (1). In our former work, we showed that super-resolution information can be obtained from the convolution of the imaging system with 3×3 or 5×5 spot arrays in numerical simulations when the NA of the illumination system equals that of the imaging objective. We wondered whether the process for $N\times N$ super-resolution spot arrays was the same as that for 3×3 or 5×5 spot arrays after the convolution. Moreover, we wondered whether the position and intensity of the center spot (after normalization) remained constant. To emphasize the generality of our approaches, we added Sections 1.2 and 1.3 to the Supplementary Notes regarding these points, demonstrating that 10×10 or 50×50 spot arrays enable super-resolution imaging. Please see pages S4-S7, lines 43-102 for more information.

We have also modified the sentence to *'To demonstrate that the super-resolution information can also be obtained when the number of spots is extended to N^2 ($N \in \mathbb{N}^*$), we first mathematically proved that $N\times N$ super-resolution spot array illumination enables super-resolution imaging (the distance between the adjacent spot center is twice the size of the spot) (see*

Supplementary Note 1).’ Please see page 5, lines 7-11.

Q11. Parameters in equation (3) are not explained, such as : f_{L1} , $f_{u,min}$, $L_{u,max}$ and Delta p . I'm assuming they are related to the SLM dimensions and pitch. Same for equation (4) - some parameters are not defined.

Reply:

We have now added the descriptions of parameters in Eqs. (3) and (4):

‘where $f_{u,min}$ is the minimum frequency of the tilted phase term, f_{L1} is the focal length of lens L1 (Fig. S10), Δp is the pixel size of the SLM, λ is the wavelength of the incident beam, and M_{sys} is the magnification of the system. The minimum step size is $\delta_{u,min} = 79.1$ nm when $L_{u,max} = 1920$ and $\lambda_1 = 405$ nm.’ and ‘The maximum spatial frequency variation in the tilted phase term can be expressed as $f_{u,max} = 2 / (L_{u,min} \Delta p)$, and the maximum scan range in the phase-shift scanning process can be expressed as’ Please see page 8, lines 1-4, 5-7, respectively.

We also checked all the equations in the manuscript and supplement materials as:

‘The typical choice for the initial complex amplitude in the focal plane is $A_0(x,y) = T_0(x,y) \exp[j\varphi_{spot}^{(0)}(x,y)]$, with an initial amplitude distribution of $T_0(x,y)$ and a random phase of $\varphi_{spot}^{(0)}(x,y)$, where $j = \sqrt{-1}$.’ and ‘After combining the above mentioned phase distributions, we added the phases of the spot array $\varphi_{spot}(x,y)$ and phase-shift scans $\varphi_{ps}(x,y)$ to the phase distribution uploaded to the phase-only SLM.’ Please see pages 16 and 18, lines 13-15, 16-18, respectively.

Q12. How is the phase shift scanning implemented? It sounds like the array has to scan the local area first before moving beyond its area.

Reply:

Phase shift scanning can be realized by varying the tilted phase. This tilted phase can be regarded as an optical wedge to implement the phase shift. The optical wedge is equivalent to an Echelon grating, and the phase distribution after discretization can be represented by the phase-shift scan $\varphi_{ps}(x,y)$. After combining the two phase distributions mentioned above, we added the phases of spot array $\varphi_{spot}(x,y)$ and phase-shift scan $\varphi_{ps}(x,y)$ to the phase distribution uploaded to the phase-only SLM. A detailed analysis of the phase-shift scanning process (theory, characteristics, maximum and minimum scanning range) is presented in Supplementary Note 2 and pages 15-17, parts

B and C.

Following the referee's suggestion, we have unified 'phase-shifting scanning', 'phase-shifting', and 'phase-shift' as 'phase-shift scanning' in our revised manuscript.

Q13. 'Uploading six stacks phase distribution on the SLMs, we respectively reconstructed super-resolution images illuminated by 10×10 (N=10) and 50×50 (N=50) spot arrays with a resolution of 0.52Airy and 0.68Airy illumination, which contained 6 images in 2 FoVs with three-wavelength incidence (1=λ405nm, 2=λ488nm, and 3=λ561nm).' it is unclear to me what the stacks are referring to. Is it 3 colors for 2 different Airy illumination patterns? Are the stacks phases on the SLM?

Reply:

We thank the reviewer for his or her constructive feedback. The term 'stack' originally referred to a series of images acquired at different focus distances in the considered area to create the final image with a greater depth of field. Similarly, in the manuscript, a stack refers to the combination of phases formed by an illumination spot array for a single incident wavelength. Therefore, for the three incident wavelengths of $\lambda_1=405$ nm, $\lambda_2=488$ nm, and $\lambda_3=561$ nm, a total of 6 sets of phases were applied to the SLM to reconstruct the nucleus, cytoskeleton, and mitochondria.

Following the reviewer's suggestions, we have modified the relevant statement to 'uploading six sets of phase distributions to the SLMs', 'the captured images', and 'sets of phase distributions. Please see pages 9, 9, 13, lines 11, 14, 11 , respectively.

As noted by the reviewer, figures 4(a) and 4(b) show 2 different Airy illumination patterns with 3 colors. Figures 4(a) and 4(b) show reconstructed superresolution images illuminated by 10×10 (N=10) and 50×50 (N=50) spot arrays with resolutions of 0.52 Airy and 0.68 Airy illumination, respectively. Please see pages 9-11 for more information.

REVIEWERS' COMMENTS

Reviewer #1 (Remarks to the Author):

The authors have answered all the questions properly, and the quality of the paper is greatly improved. I can support the publication of the paper now.

Reviewer #2 (Remarks to the Author):

Thank you to the authors for responding to the review comments. In particular, thank you for improving the overall clarity of the manuscript.

My only remaining concern is assessing the illumination properties (Q4) such as spot size, shape, and intensity uniformity across the excitation array. The authors explain that these properties are not the manuscript's focus and that the analysis would be difficult due to non-linear effects. While I understand that such an analysis might be more difficult due to the non-linear effects at play, I would have appreciated some attempt at performing such an analysis (at least intensity should be possible using a dye sample or the like). I leave this decision up to the discretion of the editor, and otherwise, I recommend the manuscript for publication.

As detailed below, we have revised our manuscript in response to the reviewers' comments. The original referee comments are shown in *blue* color, whereas for ease of communication, our answers are provided in *black* color. The changes that we have made in the manuscript text are highlighted in yellow.

Summary of our Revisions:

We have revised our manuscript according to the reviewers' comments, which will be detailed in our specific responses listed below.

We have included additional analyses of uniformity across the excitation array in the supplementary note.

Reviewer #1 (Comments to the Author):

Overall comments: *The authors have answered all the questions properly, and the quality of the paper is greatly improved. I can support the publication of the paper now.*

Reply:

We sincerely thank the reviewer for his/her affirmative comments and positive evaluations.

Reviewer #2 (Comments to the Author):

Overall comments: *Thank you to the authors for responding to the review comments. In particular, thank you for improving the overall clarity of the manuscript.*

Reply:

We appreciate the constructive feedback.

Detailed comments:

Q1. My only remaining concern is assessing the illumination properties (Q4) such as spot size, shape, and intensity uniformity across the excitation array. The authors explain that these properties are not the manuscript's focus and that the analysis would be difficult due to non-linear effects. While I understand that such an analysis might be more difficult due to the non-linear effects at play, I would have appreciated some attempt at performing such an analysis (at least intensity should be possible using a dye sample or the like). I leave this decision up to the discretion of the editor, and otherwise, I recommend the manuscript for publication.

Reply:

We agree with the reviewer's suggestions. For a point-scanning microscope, the characteristics of the spots (spot size, number, efficiency, and uniformity) are important when using multiple spots excitation. In line with the reviewer's suggestion, we have assessed the influence of uniformity across the excitation in supplementary note 5 'Analysis of uniformity across the excitation array' section as follows:

'In the mathematical demonstration part, we confirmed that the relative positions and intensities of the $N \times N$ illumination spot array are not affected by the convolution of the imaging system. However, this verification presupposes standard spot sizes, shapes and intensities. During the experimentation, efforts to enhance the uniformity of the spot array were made, yet complete uniformity was unattainable due the small spot size, as depicted in Fig. 2. Non-uniformity leads to alterations in relative positions and intensities upon convolution by the imaging

system. Here, we investigated the impact of uniformity across the excitation array in Fig. 2(a), featuring 10×10 spot array with a spot size of 0.52 Airy spot units.

Assume the intensity and position of the i_{th} spot as I_i and x_i , respectively. After convolution, the intensity and position change to I_{ci} and x_{ci} . We assessed the intensity ratio I_{ci}/I_i and position deviation represented as $\|x_{ci} - x_i\|$ for the spot array. As illustrated in Fig. S12, the intensity ratio deviation remains among $\pm 15\%$ and the position deviation is less than 15 nm. Overall, given the resolution of our imaging system, such deviations do not significantly affect the structural information of biological samples.' Please see page S25, lines 403-423.

Fig. S12. Analysis of uniformity across the excitation array. The results of **a** Intensity ratio, **b** Position deviation due to the un-uniformity across the excitation array.